# Purification and Functional Characterization of a Soluble Trehalase in *Lissorhoptrus oryzophilus* (Coleoptera: Curculionidae)

**DOI:** 10.3390/insects13100867

**Published:** 2022-09-24

**Authors:** Qingtai Wang, Kui Fang, Lizhong Qi, Xiao Wang, Yu Pan, Yunshuo Li, Jinghui Xi, Juhong Zhang

**Affiliations:** 1College of Plant Science, Jilin University, Changchun 130062, China; 2Technical Center of Kunming Customs, Kunming 650228, China

**Keywords:** *Lissorhoptrus oryzophilus*, soluble trehalase, molecular docking, RNAi, prokaryotic expression, homology modeling

## Abstract

**Simple Summary:**

The rice water weevil, *Lissorhoptrus oryzophilus* Kuschel (Coleoptera: Curculionidae), is indigenous to the United States and has become a significant invasive agricultural pest in China. In this study, we identified and cloned one trehalase gene (*LoTRE1*) encoding a soluble protein in *L. oryzophilus* and compared the relative expression levels of *LoTRE1* in different tissues. The purified *LoTRE1* protein was obtained using a prokaryotic expression system, and its enzymatic properties were explored. Amino acid sequence homology modeling of *LoTRE1* and molecular docking between the *LoTRE1* protein and substrate trehalose were simulated, which further provided a theoretical basis for revealing the role of *LoTRE1* in the degradation mechanism of trehalose. In addition, the *LoTRE1* double-stranded RNA (dsRNA) was synthesized in vitro, and its RNAi effect in *L. oryzophilus* was detected via feeding. The results suggested that *LoTRE1* played a vital role in *L. oryzophilus* development, which could be useful for providing information for insect pest control in the future.

**Abstract:**

Trehalase is the only enzyme known for the irreversible splitting of trehalose and plays a major role in insect growth and development. In this report, we describe a basic study of the trehalase gene fragment encoding a soluble trehalase from *Lissorhoptrus oryzophilus* (*LoTRE1*). Sequence alignment and phylogenetic analysis suggested that *LoTRE1* was similar to some known insect trehalases and belongs to the Coleoptera trehalase group. Additionally, *LoTRE1* was expressed mainly in the fat body. Purified protein was obtained using heterologous expression of *LoTRE1* in *Escherichia coli*, and the recombinant protein exhibited the ability to decompose trehalose. Enzyme–substrate docking indicated the potential involvement of other residues in the catalytic activity, in addition to Asp 333. Moreover, feeding of adults on *LoTRE1* dsRNA silenced the transcription of *LoTRE1* and thereby reduced the activity of trehalase and increased the trehalose content; it also led to a 12% death rate. This study reveals essential molecular features of trehalase and offers insights into the structural aspects of this enzyme, which might be related to its function. Taken together, the findings demonstrate that *LoTRE1* is indispensable for adults of this pest and provide a new target for the control of *L. oryzophilus*.

## 1. Introduction

Trehalose is a nonreducing disaccharide that consists of two α-glycosidically linked glucose units. This disaccharide is found in many organisms, such as plants, nematodes, bacteria, insects, and some other invertebrates; however, it does not exist in mammals [1]. Trehalose is mainly considered an important cell protective metabolite in vivo, and cells can synthesize trehalose in large quantities under stress and degrade it rapidly under normal conditions [2]. Trehalose exhibits multiple physiological effects in various organisms, and it has been shown that trehalose can shield proteins and cellular membranes from inactivation or denaturation caused by a range of stress conditions, including dehydration, desiccation, cold, heat, and oxidation [3,4,5,6,7,8]. Additionally, trehalose is the major blood sugar in insects, playing an important role as an instant source of energy and in the response to abiotic stresses. Trehalose is specifically synthesized in insect fat bodies and quickly discharged into the hemolymph and other tissues [7,8]. To utilize hemolymph trehalose, insect tissues contain trehalases (EC 3.2.1.28) that catalyze the hydrolysis of one mole of trehalose to two moles of glucose. Thus, trehalase is the enzyme that is required for the uptake or utilization of trehalose in the hemolymph of insects. Two types of trehalase, soluble trehalase (TRE1) and membrane-bound trehalase (TRE2), have been cloned and characterized in several insect species, such as *Acyrthosiphon pisum*, *Bombyx mori*, *Helicoverpa armigera,* and *Harmonia axyridis* [9,10,11,12]. Both TRE1 and TRE2 include a signal peptide, two signature motifs (PGGRFREFYYWDSY and QWDYPNAWPP), and one glycine-rich region [13]. Gibson et al. [14] have shown that the catalytic domain of *E. coli* trehalase displays an aspartate (Asp312) and a glutamate (Glu496) residue, which play the role of a general acid and a general base, respectively, similarly to hydrolases from the GH37 family. By site-directed mutagenesis in *Spodoptera frugiperda*, three arginine residues essential to the enzyme activity were identified inside the active site [15]. To date, no three-dimensional (3D) structure is available from experimental data for TREs from plants, animals, or fungi, whereas molecular modeling studies have predicted the 3D structure of insect TREs in *Bombyx mori* [10], *Helicoverpa armigera* [11], *Drosophila melanogaster* [16], *Chironomus riparius* [17], and *Delia antiqua* [18]. There are also individual protein differences among different insects [10,17]. In vivo, soluble trehalase accounts for the majority of the overall trehalase enzyme activity, according to previous studies [19]. The main function of TRE1 is to decompose trehalose in cells. TRE1 is an essential enzyme in insect energy metabolism and the first enzyme in the chitin synthesis pathway in insects [11].

The rice water weevil, *Lissorhoptrus*
*oryzophilus* Kuschel (Coleoptera: Curculionidae), is the most harmful and widely distributed early-season pest of rice in the USA [20] and causes serious economic issues in wetland rice agriculture, resulting in losses of up to 25% in untreated fields [21]. Parthenogenetic female *Lissorhoptrus oryzophilus* (*L. oryzophilus*) have quickly invaded many rice-growing regions around the world [20,22,23]. In China, *L. oryzophilus* were first discovered in 1998; they have rapidly invaded 78% of the provinces and have become the foremost widespread invasive pest [24]. *L. oryzophilus* have caused serious harm to the rice in the invaded area.

In this paper, we analyzed the sequence, structural characteristics, and expression patterns of *LoTRE1*. We also expressed recombinant *LoTRE1* and studied its physicochemical properties to gain insight into the optimum conditions for its activity, including temperature and pH. Furthermore, we used RNAi to study the function of the *LoTRE1* gene and detected changes in target gene expression, trehalase activity, and trehalose content. The findings of our research are critical for understanding the performance of trehalase at the molecular level, as well as the potential to adapt to future invasions. The characterization of trehalase genes could facilitate the development of novel ways to manage *L. oryzophilus*.

## 2. Materials and Methods

### 2.1. Insect Culture and Tissue Collection

*L. oryzophilus* adults were collected from Changchun, Jilin Province, China, and reared on rice seedlings under a 16 h light/8 h dark photoperiod at 26 ± 1 °C with 80 ± 5% relative humidity. The adults and different tissues (hemolymph, midgut, head, wing, fat body, and leg) of dissected *L. oryzophilus* were promptly immersed in liquid nitrogen and stored at −80 °C until use.

### 2.2. Total RNA Extraction and cDNA Synthesis

Total RNA was extracted from adults and different tissues using RNAiso Plus (Takara, Dalian, China) according to the manufacturer’s instructions for gene cloning and spatial expression. The RNA integrity and concentration were checked using agarose gel electrophoresis and spectrophotometry (NanoDrop2000, Wilmington, DE, USA), respectively. Then, approximately 1 μg of total RNA was utilized for the synthesis of first-strand cDNA by using the PrimeScript™ RT Reagent Kit with gDNA Eraser (Takara, Dalian, China). The synthesized cDNA was stored at −20 °C until use.

### 2.3. Identification of the LoTRE1 Gene and Bioinformatics Analysis

The sequence of *LoTRE1* was identified using our unpublished transcriptome data. The amino acid sequence of *LoTRE1* was deduced using DNAMAN software. Open reading frames (ORFs) of genes were predicted using ORF finder (https://www.ncbi.nlm.nih.gov/orffinder/, accessed Date: 25 June 2021). Signal peptides and transmembrane domains were predicted using SignalP-5.0 (http://www.cbs.dtu.dk/services/SignalP/, accessed Date: 25 June 2021) and TMHMM2.0 (http://www.cbs.dtu.dk/services/TMHMM/, accessed Date: 25 June 2021), respectively. The molecular weight and theoretical isoelectric point of the deduced protein were calculated using the ExPASy Compute pI/Mw tool (http://web.expasy.org/compute_pi/, accessed Date: 25 June 2021) [25]. Multiple amino acid sequence alignments were performed by using DNAMAN software (http://www.lynnon.com/pc/alignm.html, accessed Date: 25 June 2021). The tertiary structures of *LoTRE1* proteins were predicted using SOPMA (https://npsa-prabi.ibcp.fr/cgi-bin/secpred_sopma.pl, accessed Date: 25 June 2021) and SWISS-MODEL (https://swissmodel.expasy.org/, accessed Date: 25 June 2021). PDB files and molecular ligand data were obtained from ZINC (http://zinc15.docking.org/substances/home/, accessed Date: 25 June 2021). The molecular model docking calculations of *LoTRE1* with ligands were performed by using the Autodock 4.2 program. BLASTX best hits were found using the BLASTX program provided by NCBI (http://blast.ncbi.nlm.nih.gov/Blast.cgi, accessed Date: 25 June 2021). Phylogenetic trees were constructed using MEGA 6.0 software with maximum-likelihood phylogenetic analysis. The tree was colored and arranged using iTOL (https://itol.embl.de/upload.cgi, accessed Date: 25 June 2021).

### 2.4. Quantitative Real-Time PCR

Primer pairs for qPCR were designed using Primer 5 software, as shown in Table 1. GAPDH was used as a reference gene. mRNA expression levels were measured using qPCR using SYBR qPCR SuperMix (TransGen, Beijing, China). Each amplification reaction was carried out in a total volume of 20 μL, with 1 µL of cDNA, 10 µL of green qPCR SuperMix, 0.4 µL of forward primer, 0.4 µL of reverse primer, and 8.2 µL RNase-free water. qPCR was performed on an ABI 7500 Real-Time PCR System (Applied Biosystems, Carlsbad, CA, USA) under the following conditions: initial denaturation at 94 °C for 30 s, followed by 40 cycles of denaturation at 94 °C for 5 s, annealing at 55 °C for 15 s, and extension at 72 °C for 10 s. To calculate the relative expression levels, melting curves were evaluated to confirm the single peak and check amplification specificity after qPCR. Standard deviations and means were obtained from the mean of three biological replicates with three corresponding technical replicates. The relative expression value of the *LoTRE1* gene was calculated using the 2^−ΔΔCt^ method [26].

### 2.5. Protein Expression and Purification

The coding region of the *LoTRE1* gene was subcloned into the *BamHI/XhoI* sites of the pET28a (+) vector and then transformed into Rosetta (DE3) *E. coli* competent cells. The colonies were grown on Luria–Bertani culture medium with kanamycin (50 mg/mL). The positive monoclones were cultured in liquid LB medium (supplemented with 50 mg/mL kanamycin) overnight at 37 °C. The culture was diluted at 1:100 in liquid LB and incubated at 37 °C for 3–4 h until the OD600 reached 0.4–0.6. Isopropyl-β-d-thiogalactoside (IPTG) was added at final concentrations of 0.1, 0.4, 0.8, and 1.0 mmol/L, and then the culture was incubated at different temperatures (28 and 37 °C) for 8 h. Centrifugation (5000 rpm, 5 min, 4 °C) was performed to harvest the cells, and the collected cells were suspended in 1× phosphate-buffered saline (PBS). The suspension was sonicated on ice and centrifuged (5000 rpm, 5 min, 4 °C) for a second time. Protein present in the supernatant was purified with a Ni-NTA Gravity Column (Sangon, Shanghai, China). The purified protein was assessed using SDS–PAGE and then quantified via the Bradford assay with BSA as the standard.

### 2.6. Western Blot Analysis

The purified protein was separated using 10% SDS–PAGE and then transferred to a polyvinylidene fluoride (PVDF) membrane (100 V, 1 h). The membrane was treated with 5% blocking protein powder in TBST (1 M Tris-HCl pH 7.5, 500 mM NaCl, and 0.2% Tween 20) for 1 h at room temperature and reacted with the anti-6 × His tag mouse monoclonal antibody (Sangon, Shanghai, China) in TBST for 1.5 h at room temperature. The membrane was washed with TBST 4 times for 5 min each time and then incubated with AP-conjugated goat anti-mouse IgG (Sangon, Shanghai, China) in TBST for 1 h at room temperature. The NBT/BCIP substrate solution (Sangon, Shanghai, China) was utilized to visualize the protein band after the PVDF membrane was washed again.

### 2.7. Enzyme Activity Assay In Vitro

The 3,5-dinitrosalicylic acid technique was used to indirectly measure trehalase activity [27]. The reaction mixture (1 mL) consisted of 10 µL purified protein, 50 µL trehalose (200 mmol/L), and 940 µL PBS. PBS was prepared for a pH range of 3.0–7.0 The mixture was incubated in each pH buffer at 25 °C for 30 min and subjected to boiling for 5 min. The coagulated protein was removed by centrifugation at 12,000 rpm for 10 min at 4 °C. Trehalase activity was determined by measuring the content of glucose released during incubation. Similarly, for the measurement of trehalase activity at different temperatures, the mixture containing PBS (pH 7.0) and other components was incubated at the 9 individual temperatures, which ranged from 5 to 75 °C. To determine the kinetic parameters (*K_m_* and *V_max_*) of trehalase, substrates at different concentrations (1, 2.5, 5, 7.5, and 10 mmol/L) were added to the reaction mixture and incubated at 50 °C and pH 7.0 for 30 min. The trehalase activity was recorded with a UV-2450 spectrophotometer (Shimadzu, Japan). One unit of enzyme activity (U) was defined as the amount of protein that released 1 μmol of glucose in 1 min. Each experiment was replicated three times.

### 2.8. dsRNA Synthesis and Feeding

The dsRNA of *LoTRE1* was synthesized using the T7 RioMAX Express RNAi System (Promega, San Luis Obispo, CA, USA) according to the manufacturer’s recommendations. Green fluorescent protein (GFP) dsRNA was used as the control. The primers used to synthesize the dsRNA are listed in Table 1.

A total of 270 *L. oryzophilus* adults were divided into three groups, and each group was fed rice leaves with dsGFP, ds*LoTRE1*, and RNase-free water. Approximately 30 individuals were fed per treatment. Each group contained three biological replicates. Edges of approximately 1 cm were cut from both ends of fresh rice leaves, and the leaves were dried for 30 s at 55 °C in a drying oven. The rice leaves were immersed in 10 mL centrifuge tubes containing 500 ng/µL dsRNA for 6 h. The adults of *L. oryzophilus* were transferred to a new 10 mL centrifuge tube, and each tube contained 30 insects. The treated rice leaves were put into the centrifuge tubes containing *L. oryzophilus* adults, and then the centrifuge tubes were sealed with gauze. dsRNA feeding was continued for 12 h, and the leaves were replaced with fresh rice leaves without dsRNA every 24 h. The dead insects were picked out with a brush, and the number of dead insects was recorded. Fresh leaves and dsGFP-treated leaves were used as controls. Total RNA from treated *L. oryzophilus* adults was extracted as a template, and specific primers (Table 1) were used for quantitative RT–qPCR. Each RT–qPCR biological replicate contained three surviving *L. oryzophilus* adults. Three biological replicates and three technical replicates were set for each treatment group.

### 2.9. Determination of Trehalase Activity and Sugar Content In Vivo

A total of 30 *L. oryzophilus* adults were placed in a 10 mL centrifuge tube and fed with rice leaves treated with dsRNA for 12 h, then transferred to fresh rice leaves without dsRNA. The surviving individuals were picked out with a brush after 48 h. The enzymatic analysis of the activity of trehalase (THL) was performed using a THL kit according to the manufacturer’s instructions (Solarbio, Beijing, China), and the trehalose content was determined according to the manufacturer’s instructions for the Trehalose Content Kit (Solarbio, Beijing, China). Each experiment was replicated three times.

### 2.10. Statistical Analysis

All data obtained during this study were expressed as the means ± standard deviations of three replicates and were tested with a one-way ANOVA and *t* tests using SPSS 22.0. *p* values of less than 0.01 indicated significant (**) differences.

## 3. Results

### 3.1. Sequence Analysis of LoTRE1

LoTRE1 was identified from our unpublished transcriptome data of *L. oryzophilus*. Through homology searching in our transcriptional sets, the sequence of *LoTRE1* was found to contain an 1839 bp-long open reading frame encoding 612 amino acid residues (approximately 70833 Da and a theoretical isoelectric point (pI) of 5.38). A signal peptide of 25 amino acids and a cleavage site (IAI-YK) between residues 25 and 26 were identified (Figure 1). *LoTRE1* had two signature motifs (PGGRFREFYYWDSY and QWDYPNAWPP) and a highly conserved glycine-rich region (GGGGEY). Four N-glycosylation sites (residues 336, 579, 580, and 581) were identified in the *LoTRE1* sequence (Figure 1). TMHMM predicted that there were no transmembrane domains.

Multiple sequence alignment was performed to analyze evolutionarily or structurally related positions between *LoTRE1* and its homologues based on the amino acid sequence (Figure 2). *LoTRE1* showed approximately 55–72% sequence identity with other Coleoptera insect trehalases. The greatest sequence identity (72%) was with trehalase (GenBank ID: XP 030756586.1) from *Sitophilus oryzae,* and the lowest sequence identity (55%) was with trehalase (GenBank ID: XP_031336857.1) from *Photinus pyralis.* To investigate the evolutionary relationships of *LoTRE1*, a phylogenetic tree of *LoTRE1* was constructed along with several previously studied TREs from 24 species of Coleoptera, Hemiptera, Blattaria, and Hymenoptera using maximum-likelihood phylogenetic analysis (Figure 3). The results showed that *LoTRE1* shared the highest homology with the genes from *Sitophilus oryzae* and *Rhynchophorus ferrugineus.*

### 3.2. Spatial Expression Patterns of LoTRE1

The expression profiles of *LoTRE1* in numerous tissues were probed using qPCR. *LoTRE1* was shown to be expressed in a variety of tissues, including the head, midgut, hemolymph, wing, leg, and fat body (Figure 4). Remarkably, *LoTRE1* had high expression within the fat body; however, it had the lowest expression in the wing. The relative expression in the fat body was 120 times that in the wing.

### 3.3. Molecular Docking and Key Binding Sites of LoTRE1

The SWISS-MODEL service was used to predict the three-dimensional structure of *LoTRE1* based on the resolved crystal structure of the periplasmic trehalase of *E. coli* (PDB ID 2WYN; Figure 5). A Ramachandran plot and the Procheck server were used to verify the homology model’s dependability (Figure 6). A total of 89.7% of amino acid residues were in the most favored regions, 9.9% of amino acid residues were in the additional regions, and 0.2% of amino acid residues were in the disallowed regions. Thus, this was a high-quality model. The predicted α-toroidal structure of *LoTRE1* was very similar to that of the periplasmic trehalase of *E. coli*, but one β-sheet was absent in *LoTRE1* at position 1.

A molecular docking study was performed to further understand the interaction of trehalose with trehalase and to obtain insight into the binding location of this molecule on trehalase. The 3D protein structure with small molecular compounds was studied using molecular docking simulations (Figure 7). The results showed that trehalose bound tightly to *LoTRE1* (G = −1.36 kcal/mol). The docking results revealed that the interactions of trehalose with *LoTRE1* were mediated by ARG 179, TRP 186, ASN 223, ARG 232, GLN 234, ARG 297, GLU 299, GLY331, ASP 333, and TRP 477.

### 3.4. Protein Expression and Purification

To obtain recombinant *LoTRE1* protein in large quantities, we induced protein production in *E. coli* Rosetta (DE3) cells under different induction conditions (Figure 8A). The molecular weight of the trehalase protein was found to be close to the predicted size in all the tests (70 kDa). Therefore, we chose a final concentration of 0.1 mmol/L IPTG at 28 °C to obtain large quantities of recombinant protein. In addition, the *LoTRE1* recombinant protein was mostly expressed as soluble protein (Figure 8B). The purified *LoTRE1* protein was analyzed using SDS–PAGE (Figure 8C), and one obvious unique band appeared at the predicted size (70 kDa). The expression of the recombinant *LoTRE1* protein was verified using Western blotting (Figure 8D). The concentration of the purified *LoTRE1* protein was 0.98 mg/mL.

### 3.5. Enzymatic Assays of LoTRE1 In Vitro

The enzyme activity increased with increasing temperature and pH. The optimum activity of *LoTRE1* was found at 50 °C and pH 7.0, beyond which the enzyme activity decreased. No catalytic ability was found at temperatures above 75 °C or below 5 °C, indicating that the catalytic performance of *LoTRE1* had a restricted temperature range. Under the optimal conditions, the specific activity of purified *LoTRE1* was 58.47 ± 1.74 U/mg. The kinetic parameters *km* and *Vmax* of *LoTRE1* were 48.6 mmol/L and 1.108 mmol/(L·min), respectively (Figure 9).

### 3.6. Silencing of LoTRE1 by RNAi

qPCR was performed to prove the effectiveness of RNAi in *L. oryzophilus*. Feeding dsRNA-*LoTRE1* significantly reduced the expression levels of trehalase in the rice water weevil compared with that in the nontarget control group within 24 and 48 h (*p* < 0.01). The results showed that RNAi reduced the expression levels of *LoTRE1* to 40 % at 24 h and 56 % at 48 h in *L. oryzophilus* (Figure 10A). The mortality rate of *L. oryzophilus* increased significantly (*p* < 0.01) at 24 h (9%) and 48 h (12%) after dsTRE1 feeding (Figure 10B).

### 3.7. The Effect of LoTRE1 Silencing on Trehalose Metabolism

At 48 h after feeding dsRNA-*LoTRE1*, the trehalase activity of *L. oryzophilus* was 13.14 ± 0.47 U/g, which was 31.6% lower than that of the control group (Figure 11A), and there was a significant difference compared with the control group (*p* < 0.01). The trehalose content of *L. oryzophilus* was 5.73 ± 0.60 mg/g, which was 130% higher than that of the control group (Figure 11B), and there was a significant difference compared with the control group (*p* < 0.01).

## 4. Discussion

Trehalase catalyzes the hydrolysis of trehalose and plays a vital role in insect metabolism. The amount of trehalases has been found to differ among insect species. A total of three trehalase genes were found in *N. lugens*, and three trehalase genes were identified in *Tribolium castaneum* [28,29]. One trehalase gene, *LoTRE1*, was identified from our previous transcriptome database of *L. oryzophilus*, and the analysis of the deduced amino acid sequence demonstrated that *LoTRE1* encodes a soluble trehalase. *LoTRE1* contained some conserved regions, including one signal peptide, two signature motifs, four putative glycosylation sites, and a highly conserved glycine-rich sequence. The findings were consistent with those for trehalase genes found in other Coleoptera species. Amino acid sequence alignment showed that *LoTRE1* shared high identity with some known insect trehalases. The *LoTRE1* sequence shared the highest identity with that of *S. oryzae* (72% identity GenBank ID: XP 030756586.1), followed by those of *R. ferrugineus* (70% identity GenBank ID: KAF7271780.1), *A. planipennis* (65% identity GenBank ID: XP_018322659.1), *H. axyridis* (59% identity GenBank ID: AOT82130.1), and *P. pyralis* (55% identity GenBank ID: XP_031336857.1). The phylogenetic tree revealed that *LoTRE1* shared the highest homology with the genes from *Sitophilus oryzae* and *Rhynchophorus ferrugineus.*

The model of *LoTRE1* shared a general α-toroidal architecture with the template protein. According to the CAZy database, insect trehalases are mostly members of GH family 37 and possess glutamic acid (Glu) as a nucleophile, while their proton donor is aspartic acid (Asp) [16]. We identified Asp 333 and Glu 299 as important catalytic residues in the *LoTRE1* model by comparing it to the *E. coli* periplasmic trehalase. The trehalose molecule occupies a space in the active center pocket of the enzyme that contains potential acid (Asp 333) and base (Glu 299) residues, as well as three conserved Arg residues (R 179, R 232, R 297). Our results with *LoTRE1* were consistent with those reported in other insect trehalases [9,30]. The primary three-dimensional structure of *LoTRE1* was similar to the periplasmic trehalase of *E. coli.* Both structures mainly consisted of α-helixes and were surrounded by α-toroidal structures. This structure was also observed in the trehalase of other insects [9,17]. According to Silva et al., guanidine groups and Arg residues are required for soluble trehalase activity, and chemical alteration of these residues results in enzyme inactivation. This was later confirmed by site-directed alterations in three Arg residues within the enzyme’s active site pocket [15].

In some insect species, TRE1 has been purified from the goblet cell cavity, hemolymph, egg homogenates, and midgut [31]. The tissue expression pattern analysis revealed that *LoTRE1* was most highly expressed in the fat body, followed by the midgut. Yu et al. (2021) found that trehalase was most highly expressed in the head and wings of *Diaphorina citr* [32]. Ma et al. (2015) found the highest expression of trehalase in the midgut in *Helicoverpa armigera* [33]. These results indicate that TRE1 could serve completely different functions in several tissues during the development of various insects. The insect fat body functions as an energy storage center. It is also the organization center of the metabolic processes of insects, such as growth, development, metamorphosis, and reproduction. Trehalase could cooperate with the hormones in fat to dynamically control the concentrations of trehalose and glucose in insects [34].

Trehalases can be divided into acidic trehalases and neutral trehalases according to their optimal pH. Inagaki et al. (2001) found that the optimum pH of the trehalase from *Acidobacterium capsulatum* was 2.5 [35]. Lee et al. (2001) revealed that the optimum pH of the trehalase from *Apis mellifera* was 6.7 [36]. In this study, the recombinant *LoTRE1* protein had enzyme activity in the range of pH 3.0~9.0, but the highest activity was observed at pH 7.0, which indicated that the *LoTRE1* protein has the best enzyme activity in neutral environments. Previous studies have shown that the optimum temperature of trehalase is generally high, usually at 40~65 °C [37]. In this experiment, *LoTRE1* had no activity at temperatures below 5 °C, and the enzyme activity increased gradually with increasing temperature. When the system temperature exceeded 75 °C, *LoTRE1* activity was lost. Shukla et al. (2016) revealed that the optimum temperature of the trehalase from *Drosophila melanogaster* was 55 °C [16], Ai et al. (2018) revealed that the optimum temperature of the trehalase from *Helicoverpa armigera* was 55 °C [11]. Since trehalase remains the only enzyme known for the irreversible splitting of trehalose under physiological conditions, heterologous expression of *LoTRE1* can be used to explore the biochemical characteristics and kinetic properties of the enzyme. 

RNAi is a highly conserved mechanism initiated by sequence-specific double-stranded RNA (dsRNA), leading to target-specific endogenous gene silencing [38]. RNAi has been widely accepted as a powerful tool for gene function research and is relatively well-established in a variety of insects, such as *Bemisia tabacii* [39] and Hemiptera [40]. Previous studies have shown that silencing trehalase in *Leptinotarsa decemlineata* caused larval death [32]. Silencing of trehalase in *Nilaparvata lugens* caused phenotypic deformities [41]. These results suggested that trehalase has a biological function in insect development and survival. In this study, RNAi was performed to determine whether the levels of *LoTRE1* would affect *L. oryzophilus* development and survival. The results indicated that dsRNA feeding-mediated silencing of the *LoTRE1* gene caused not only downregulation of the transcript level of *LoTRE1* but also decreased trehalase activity in treated adults. In contrast, silencing of *LoTRE1* increased the trehalose content and resulted in lethal effects in adults. In agreement with our results, knockdown of the trehalase gene led to mortality in treated *Harmonia axyridis* and *Spodoptera exigua* [12,42]. In addition, silencing HaTRE1 led to significant decreases in the ability of females to attract males and successful mating proportions [43]. Trehalose has been well demonstrated in insect physiology as an energy source for insects, maintaining the glucose level [44]. The energy level and blood glucose content are affected in various cells of insects when the hydrolysis of trehalose to glucose is inhibited, and thus other physiological pathways are affected [42,45]. Trehalase can decompose the important energy storage material and the stress metabolite trehalose in insects. The changes in gene expression and enzyme activity affect the life processes of insects, including molting, metamorphosis, and reproduction [34]. Studies have shown that the TRE gene affects the levels of three sugars by regulating gene expression and enzyme activity [44,46]. These results collectively indicated that the *LoTRE1* gene plays a vital role in *L. oryzophilus* by regulating the trehalose content. Feeding with dsTRE1 can disrupt the metabolism of trehalose in the body. The results lay a foundation for exploring the potential functions and regulatory mechanisms of insect TRE1, which could be useful for providing information for insect pest control in the future.

In conclusion, we identified one soluble trehalase gene from *L. oryzophilus*, analyzed its molecular characteristics, and explored its biochemical characteristics, kinetic properties, and optimum reaction conditions through heterologous expression analysis. The importance of *LoTRE1* was inferred through RNAi. *LoTRE1* is a key gene regulating the expression of trehalase and the trehalose content. However, the particular role of this gene is still unknown, and more research is needed to gain better knowledge of *LoTRE1’s* functions. The results of this study provide a good basis for further studies on the regulation of the expression of this gene and provide a potential target for the control of *L. oryzophilus* in the field.

## Figures and Tables

**Figure 1 insects-13-00867-f001:**
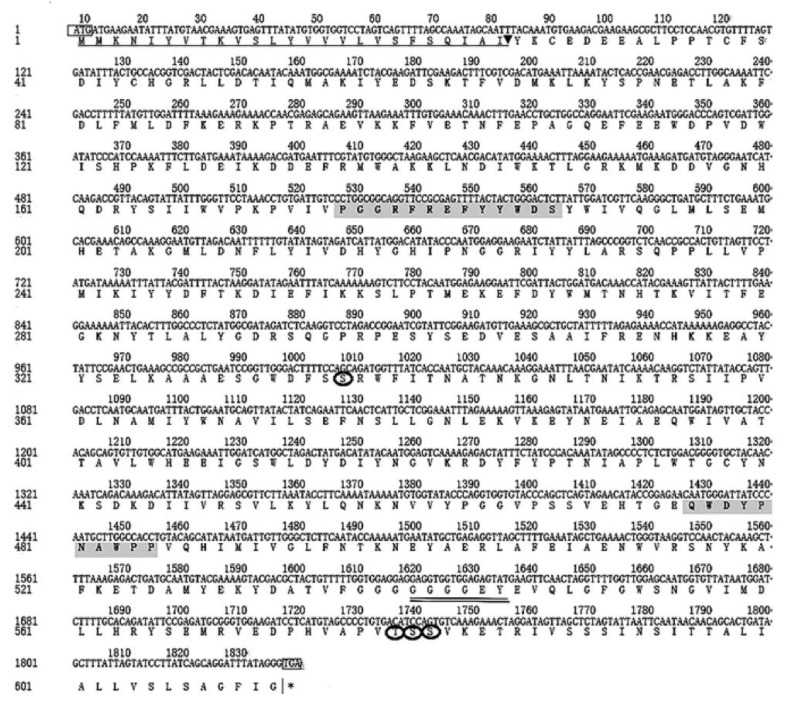
Nucleotide sequence and deduced amino acid sequence of *LoTRE1*. The black box in the figure indicates the start codon (ATG) and stop codon (TGA). Underlined amino acid residues (1–25) and the arrowhead represent the signal peptide and putative cleavage site, respectively. Trehalase signature motifs (amino acid residues 176–189 and 476–485) are shaded. Four potential N-glycosylation sites (amino acid residues 336, 579, 580, and 581) are encircled. The highly conserved glycine-rich region is double underlined.

**Figure 2 insects-13-00867-f002:**
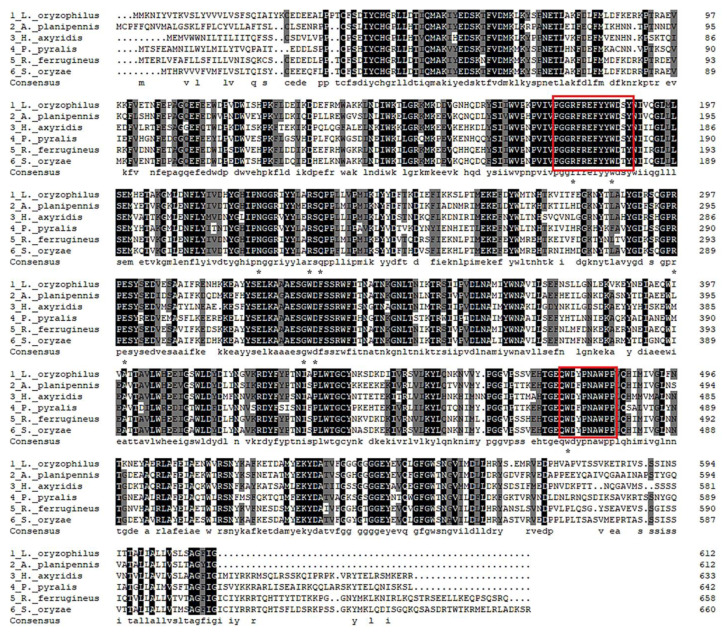
Multiple alignment of soluble trehalase amino acid sequences from Coleoptera insects. The following sequences were utilized in the alignment: *ApTRE* from *Agrilus planipennis* (XP_018322659.1), *HaTRE* from *Harmonia axyridis* (AOT82130.1), *PpTRE* from Photinus pyralis (XP_031336857.1), *RfTRE* from *Rhynchophorus ferrugineus* (KAF7271780.1), and *SoTRE* from *Sitophilus oryzae* (XP 030756586.1). Alignments were performed with ClustalX2. The interaction sites between trehalose and *LoTRE1* are indicated by (*). The two signature motifs based on *Lissorhoptrus oryzophilus* TRE1 are in red boxes in the alignment. Conserved and highly conserved amino acid residues are highlighted in grey and black, respectively.

**Figure 3 insects-13-00867-f003:**
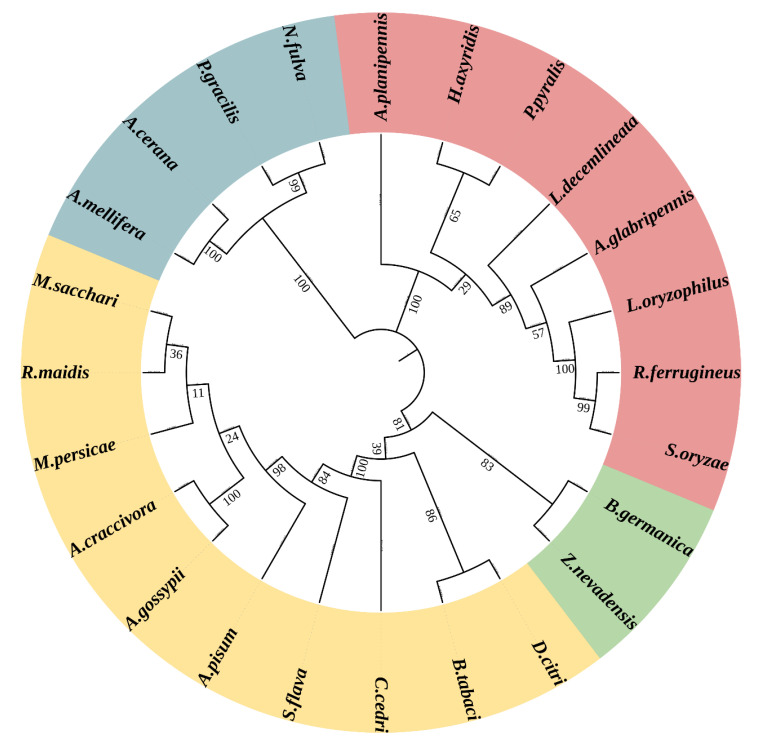
The phylogenetic tree was constructed using the maximum-likelihood method based on amino acid sequence alignment. Full-length amino acid sequences were aligned using the Mega 6 program to generate a phylogenetic tree. A bootstrap analysis was carried out, and the robustness of each cluster was verified in 1000 replications. Red represents Coleoptera, brown represents Hemiptera, green represents Blattaria, and blue represents Hymenoptera. *A. mellifera*, NP_001106141.1; *A. cerana*, XP_016903816.1; *P. gracilis*, XP_020290660.1; *N. fulva*, XP_029165580.1; *A. planipennis*, XP_018322659.1; *H. axyridis*, AOT82130.1; *P. pyralis*, XP_031336857.1; *L. decemlineata*, XP_023020910.1; *A. glabripennis*, XP_018571289.1; *R. ferrugineus*, KAF7271780.1; *S. oryzae*, XP 030756586.1; *B. germanica*, PSN49112.1; *Z. nevadensis*, KDR17472.1; *D. citri*, P_008474901.1; *B. tabaci*, XP_018905428.1; *C. cedri*, VVC30114.1; *S. flava*, *XP_025410170.1*; *A. pisum*, XP_003248025.1; *A. craccivora*, KAF0772952.1; *M. persicae*, XP_022174955.1; *R. maidis*, XP_026821537.1; *M. sacchari*, XP_025190889.1.

**Figure 4 insects-13-00867-f004:**
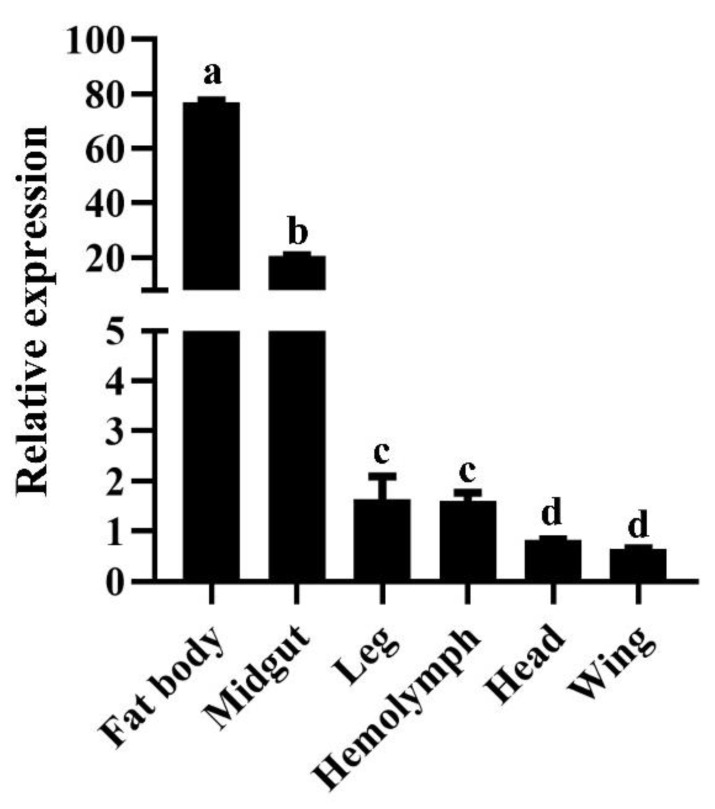
Relative expression levels of *LoTRE1* in different tissues of adult *L. oryzophilus*. All data obtained were expressed as the means ± standard deviations of three replicates and were tested using a one-way ANOVA and *t* tests using SPSS 22.0. The significant changes are indicated by different letters above the bars. (*p* < 0.05).

**Figure 5 insects-13-00867-f005:**
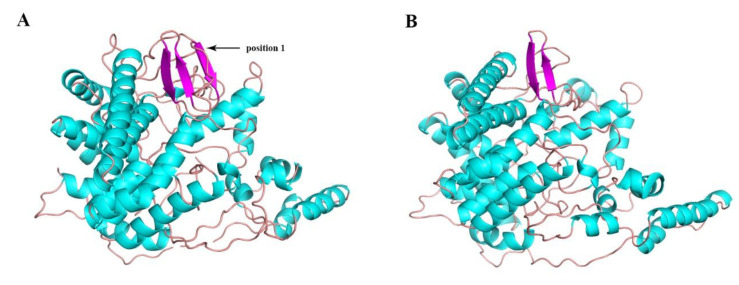
Three-dimensional structure of the *E. coli* periplasmic trehalase (**A**) and the homology model of the *L. oryzophilus* trehalase (**B**). The differences between the structures are highlighted with arrows.

**Figure 6 insects-13-00867-f006:**
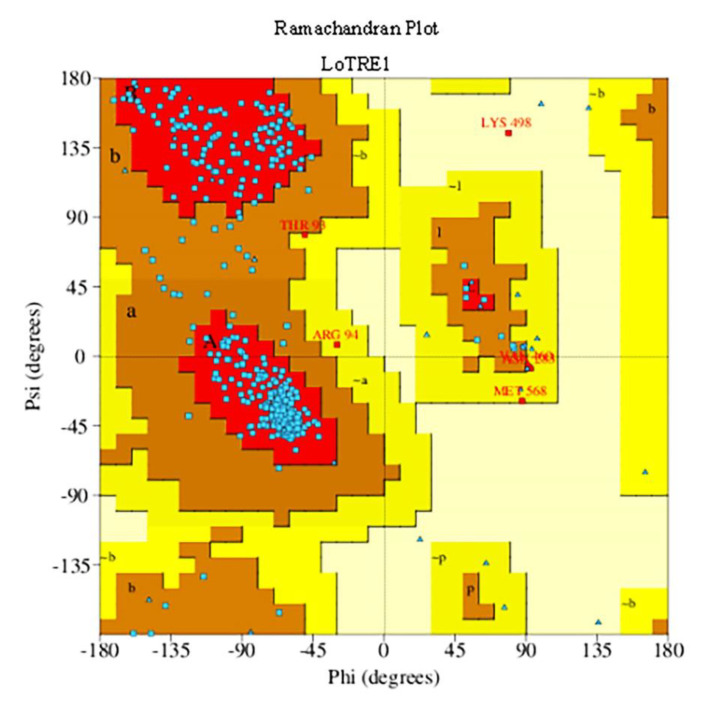
Ramachandran plot of *LoTRE1*.

**Figure 7 insects-13-00867-f007:**
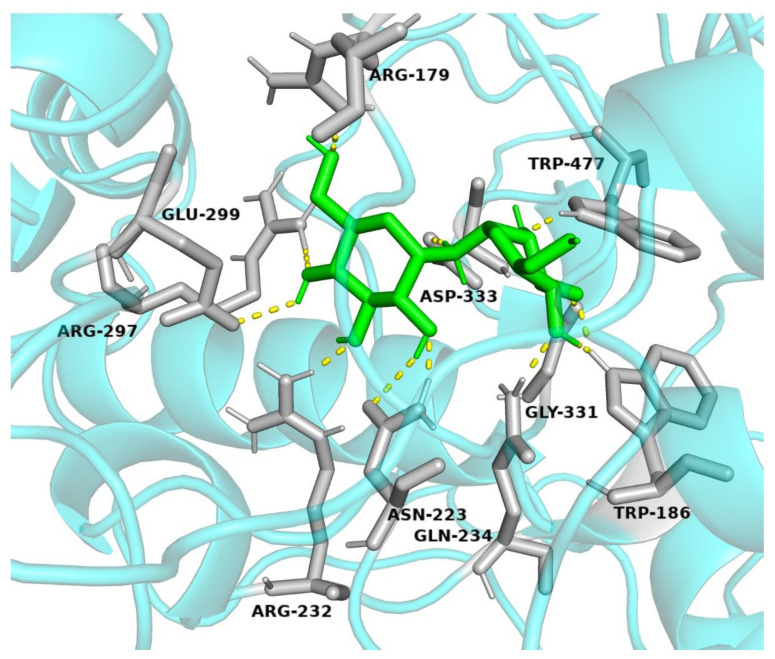
Molecular docking of *LoTRE1*. The ligands and the key residues are shown as stick models. Hydrogen bonds are shown as yellow lines, ligands are shown in green, and key residues are shown in grey. The figure was drawn using PyMOL.

**Figure 8 insects-13-00867-f008:**
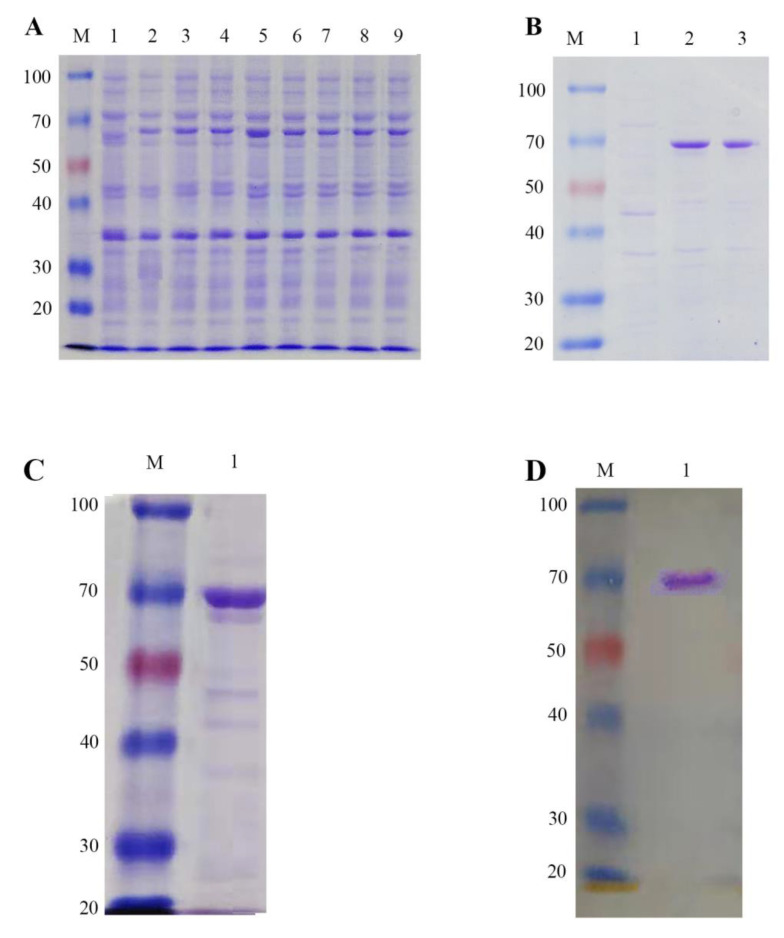
The recombinant protein *LoTRE1* was expressed and purified. (**A**) Expression of recombinant *LoTRE1* protein induced with IPTG under various conditions. Lane M: molecular weight marker. Lane 1: Uninduced cells of *E. coli* Rosetta (DE3). Lanes 2–4: The protein production in the *E. coli* Rosetta (DE3) cells induced with 0.1, 0.4, 0.8, and 1.0 mmol/L IPTG at 28 °C. Lanes 5–9: Protein production in the *E. coli* Rosetta (DE3) cells induced with 0.1, 0.4, 0.8, and 1.0 mmol/L IPTG at 37 °C. (**B**) Determination of the target protein’s expression form. Lane M: molecular weight marker. Lane 1: Uninduced cells of *E. coli* Rosetta (DE3). Lane 2: The target protein in the supernatant after sonication and centrifugation. Lane 3: The target protein in the sediments after sonication and centrifugation. (**C**) Purification of the *LoTRE1* recombinant protein. Lane M: Molecular weight marker. Lane 1: Purified target protein. (**D**) Western blot. Lane M: Molecular weight marker. Lane 1: Purified target protein.

**Figure 9 insects-13-00867-f009:**
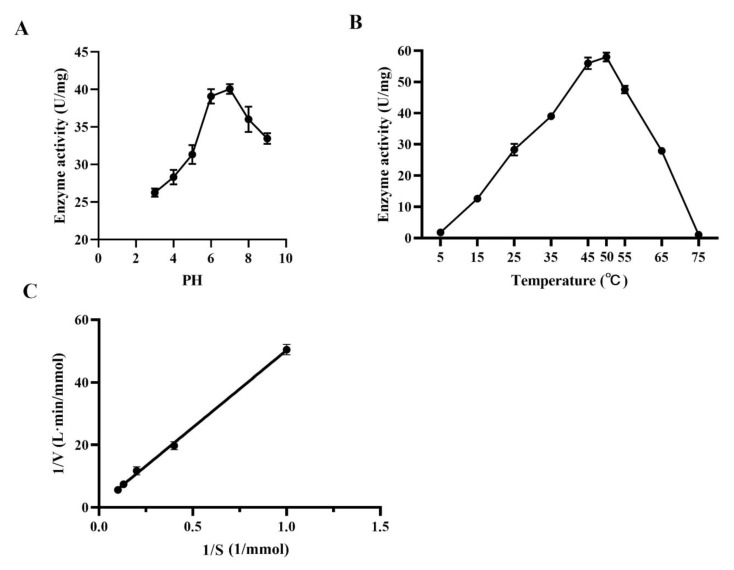
Enzymatic properties of the *LoTRE1* recombinant protein. (**A**) Enzyme activity of trehalose under different pH conditions. (**B**) Enzyme activity of trehalose under different temperature conditions. (**C**) Steady state kinetic equation of *LoTRE1*.

**Figure 10 insects-13-00867-f010:**
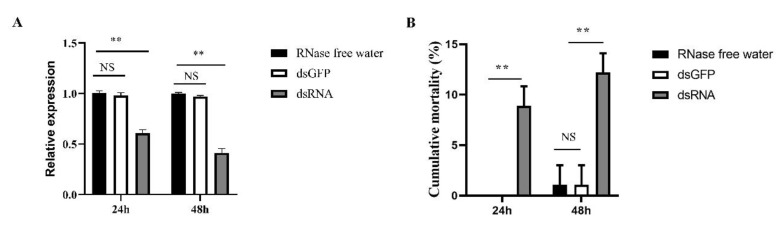
Silencing of *LoTRE1* using RNAi. (**A**) The relative expression levels of the *LoTRE1* gene after silencing for 24 and 48 h. (**B**) The mortality rates of adults after feeding on different treated leaves for 24 and 48 h. All data obtained were expressed as the means ± standard deviations of three replicates and were tested using a one-way ANOVA and *t* tests using SPSS 22.0. No significant differences are indicated by NS. The significant differences are indicated by ** (*p* < 0.01).

**Figure 11 insects-13-00867-f011:**
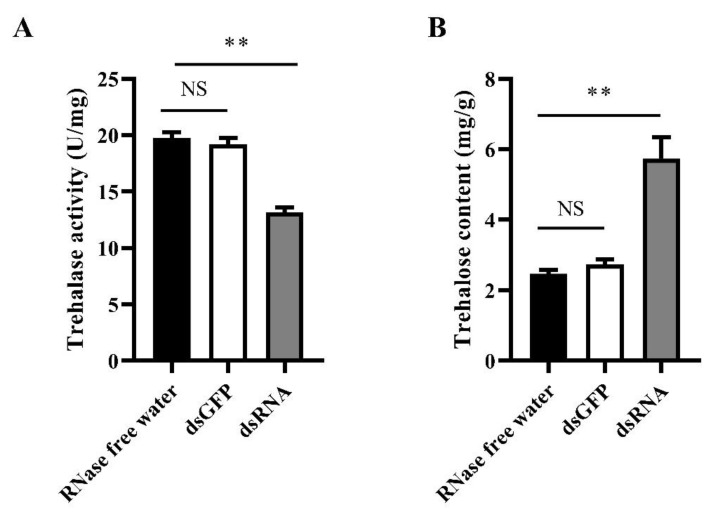
Detection of enzyme activity and trehalase content after RNAi treatment. (**A**) Trehalase activity after RNAi treatment. (**B**) Trehalose content after RNAi treatment. All data obtained were expressed as the means ± standard deviations of three replicates and were tested using a one-way ANOVA and *t* tests using SPSS 22.0. The standard deviation of the computed means based on three biological replicates is represented by the error bars. No significant differences are indicated by NS. The significant differences are indicated by ** (*p* < 0.01).

**Table 1 insects-13-00867-t001:** Primer information.

Primer Name	Primer Sequence (5′-3′)	Purpose
TRE1	F: CGCGGATCCATGATGAAGAATATTTATGTAACGA R: CCGCTCGAGTCACCCTATAAATCCTGCTGATAAG	Prokaryotic expression
dsRNA-TRE	F: TAATACACTCACTATAGGGGTGGGCTAAGAAGCTCAACG R: TAATACACTCACTATAGGGCCGAATACGATTCCGGTCTA	RNAi
dsRNA-GFP	F: TAATACGACTCACTATAGGGGACGTAAACGGCCACAAGTT R: TAATACGACTCACTATAGGGTGTTCTGCTGGTAGTGGTCG	RNAi
TRE1-q	F: AAAATTACACTTTGGCCCTCTA R: GTCCCAACCGGATTCAGC	RT–qPCR
GADPH	F: ACCACTGTCCACGCAACT R: ACTCTGAAGGCCATACCG	RT–qPCR

The boxes mark the restriction sites and the T7 promoter sequence is underlined.

## Data Availability

The data presented in this study are available in the article.

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
