# Peer review of "Purification and Functional Characterization of a Soluble Trehalase in *Lissorhoptrus oryzophilus* (Coleoptera: Curculionidae)"

_insects, 2022, doi:10.3390/insects13100867_

Round 1

Reviewer 1 Report (Previous Reviewer 3)

The authors have answered to the reviewer's points. The article is therefore acceptable for publication.

Author Response

Dear  reviewers:

Thank you for your letter and the reviewers’ comments on our manuscript entitled “Purification and functional character rization of a soluble trehalase in Lissorhoptrus oryzophilus (Coleoptera: Curculionidae)” (ID: insects-1922351). Those comments are very helpful for revising and improving our paper, as well as the important guiding significance to other research.

Thank you very much for your affirmation of this paper

Reviewer 2 Report (New Reviewer)

The authors identified LoTRE1 from transcriptome data, and LoTRE1 was further characterized via alignment, phylogenetic tree analysis and molecular docking analysis, and then expression and purification of TRE1 were performed for TRE1 activity analysis, and further RNAi further confirmed LOTRE1 plays important roles for survival through regulation of Trehalsse concentration in Lissorhoptrus oryzophilus. The experiments were well designed and performed. This study provides a potential method to control Lissorhoptrus oryzophilus. However, there are some questions to be addressed before it can be accepted for publication.

1.       Given the primers in Table 1, you expressed the full length of LoTRE1 with the signal peptide, while the signal peptide should be cut off in vivo, will the signal peptide affect your following assay? And did you remove the signal peptide in the structure analysis?

2.       The Genbank ID in figure 3 is missing, and would you like to add TRE1 in the figure legend to help readers understand them quickly? Given your phylogenetic tree, it seemed to be evaluated by bootstrap analysis, but the analysis parameter is missing.

3.       Line 62 Spodoptera frugiperda and line 68 Invivo should be italic.

4.       Line 97, how old are the adults for RNA extraction?

5.       Line 144, should be diluted at 1:100.

6.       Line 166, the purified protein concentration is missing.

7.       In 2.9. Determination of trehalase activity and sugar content in vivo, what samples you collected were used for this assay? How did you collect the samples?

8.       Please indicate the statistical analysis method in figure 4, 10 and 11.

9.       The resolution of Figure 1, 2 and 6 is low.

10.   Please add more details in figure legends such as the methods for analysis and the correct figure title.

11.   Figure5 title is not correct, A and B are from different species, so the title should be a comparison of them.

12.   For the molecular docking analysis, ARG 179, TRP 186, ASN 223, ARG 232, GLN 234, 281 ARG 297, GLU 299, GLY331, ASP 333, and TRP 477 are required to interact with trehalose, does that mean these sites are conserved? If so, please indicate these sites in figure 2.

13.   For the WB analysis, why the band of TRE1 in figure 8D is blue? Usually, the WB results show black bands, and only the prestained molecular weight marker is colorful. Please double-check your results shown here. And the band in figure 8c is wider than that of 8A, please uniform the scale.

14.   The Chinese words in line 301 and 302 should be deleted.

15.   For the RNAi part, when were the adults fed with dsRNA, day 1 or day 2?

16.   In figure 9, what’s the amount of purified TRE1 used for these analyses?

17.   Line 369, please remove the blank.

18.   Line 399, What’s the Ai in “Ai revealed”?

19.   Line 418, Ha Treh or HaTRE?

20.   Line 472, Bombyx mori should be black.

Author Response

Dear reviewers:

Thank you for your letter and the reviewers’ comments on our manuscript entitled “Purification and functional character rization of a soluble trehalase in Lissorhoptrus oryzophilus (Coleoptera: Curculionidae)” (ID: insects-1922351). Those comments are very helpful for revising and improving our paper, as well as the important guiding significance to other research. We have studied the comments carefully and made corrections which we hope meet with approval. We tried our best to improve the manuscript and made some changes in the manuscript. The main corrections are in the manuscript and the responds to the reviewers’ comments are as follows (the replies are highlighted in blue).

Responds to the reviewers’ comments:

1.Given the primers in Table 1, you expressed the full length of LoTRE1 with the signal peptide, while the signal peptide should be cut off in vivo, will the signal peptide affect your following assay? And did you remove the signal peptide in the structure analysis?

Response: Refering to the references of G von Heijne (1985, 1986), we speculate the function of signal peptide is translocation through the membrane of the endoplasmic reticulum. We agree with you that the signal peptide should be cut off in vivo. But we have consulted a great deal of relevant literatures and found that most of them did not mention that signal peptide would affect following assay (Forcella et al., 2012; Dong et al., 2021). Based on that, we did not remove the signal peptide in the structure analysis.

G von Heijne. Signal sequences. The limits of variation. J Mol Biol. 1985, 5, 184(1):99-105. doi: 10.1016/0022-2836(85)90046-4. PMID: 4032478.

G von Heijne. A new method for predicting signal sequence cleavage sites. Nucleic Acids Res. 1986, 14, 4683-4690.

Forcella M, Mozzi A, Bigi A, Parenti P, Fusi, P. Molecular cloning of soluble trehalase from Chironomus riparius larvae, its heterologous expression in escherichia coli and bioinformatic analysis. Arch. Insect Biochem. Physiol. 2012, 81, 77-89. doi:10.1002/arch.21041.

Dong C, Fan Q, Li X , et al. Expression and characterization of a novel trehalase from Microvirga sp. strain MC18[J]. Protein Expression and Purification, 2021, 182:105846.

  1. The Genbank ID in figure 3 is missing, and would you like to add TRE1 in the figure legend to help readers understand them quickly? Given your phylogenetic tree, it seemed to be evaluated by bootstrap analysis, but the analysis parameter is missing.

Response: We agree with the comment and have added the Genbank ID to help readers understand them well. The phylogenetic tree was evaluated by bootstrap analysis. And we have added the analysis parameter in the manuscript.

We re-wrote the sentence in the revised manuscript as the following: The phylogenetic tree was constructed using the maximum-likelihood method based on amino acid sequence alignment. Full-length amino acid sequences were aligned using the Mega 6 program to generate a phylogenetic tree. A bootstrap analysis was carried out, and the robustness of each cluster was verified in 1000 replications. Red represents Coleoptera, brown represents Hemiptera, green represents Blattaria, and blue represents Hymenoptera. A. mellifera, NP_001106141.1; A. cerana, XP_016903816.1; P. gracilis, XP_020290660.1; N. fulva, XP_029165580.1; A. planipennis, XP_018322659.1; H. axyridis, AOT82130.1; P. pyralis, XP_031336857.1; L. decemlineata, XP_023020910.1; A. glabripennis, XP_018571289.1; R. ferrugineus, KAF7271780.1; S. oryzae, XP_030756586.1; B. germanica, PSN49112.1; Z. nevadensis, KDR17472.1; D. citri, P_008474901.1; B. tabaci, XP_018905428.1; C. cedri, VVC30114.1; S. flava, XP_025410170.1; A. pisum, XP_003248025.1; A. craccivora, KAF0772952.1; M. persicae, XP_022174955.1; R. maidis, XP_026821537.1; M. sacchari, XP_025190889.1. Line 250-261.

  1. Line 62 Spodoptera frugiperda and line 68 Invivo should be italic.

Response: We apologize for the language problems in the original manuscript. We have changed “Spodoptera frugiperda” and line 68 “In vivo” to “Spodoptera frugiperda” and “In vivo”.

  1. Line 97, how old are the adults for RNA extraction?

Response: The overwintering adults were collected from the paddy field in June and transferred to the indoor rice seedlings to be raised. After about a week, the more active adults were selected for RNA extraction.

  1. Line 144, should be diluted at 1:100

Response: We apologize for the language problems in the original manuscript. We re-wrote the sentence in the revised manuscript as the following: The culture was diluted at 1:100 in liquid LB and incubated at 37 ℃ for 3-4 h until the OD600 reached 0.4-0.6.

  1. Line 166, the purified protein concentration is missing.

Response: We are very sorry for our negligence. We have added the sentence“The concentration of the purified LoTRE1 protein was 0.98 mg/ml”.

  1. In 2.9. Determination of trehalase activity and sugar content in vivo, what samples you collected were used for this assay? How did you collect the samples?

Response: It is our negligence and we are sorry about this. A total of 30 L. oryzophilus adults were placed in 10 ml centrifuge tube and fed with rice leaves treated by dsRNA for 12h, then transferred to fresh rice leaves without dsRNA. The survival individuals were picked out with a brush after 48 h.

  1. Please indicate the statistical analysis method in figure 4, 10 and 11.

Response: Thank you for your advice and we have added the statistical analysis method in figure 4, 10 and 11 as following: All data obtained were expressed as the means ± standard deviations of three replicates and were tested by one-way ANOVA and T tests using SPSS 22.0. The significant differences are indicated by ** (P < 0.01).

  1. The resolution of Figure 1, 2 and 6 is low.

Response: Considering the Reviewer’s suggestion, we replaced images in Figure 1, 2, 6 with higher resolution images.

  1. Please add more details in figure legends such as the methods for analysis and the correct figure title.

Response: Thank you for your advice and we have added more details in figure legends.

  1. Figure5 title is not correct, A and B are from different species, so the title should be a comparison of them.

Response: We apologize for the language problems in the original manuscript. We re-wrote the sentence in the revised manuscript as the following: Three-dimensional structure of the E. coli periplasmic trehalase (A) and the homology model of the L. oryzophilus trehalase (B). The differences between the structures are highlighted with arrows.

  1. For the molecular docking analysis, ARG 179, TRP 186, ASN 223, ARG 232, GLN 234, 281 ARG 297, GLU 299, GLY331, ASP 333, and TRP 477 are required to interact with trehalose, does that mean these sites are conserved? If so, please indicate these sites in figure 2.

Response: Yes, it is mean these sites are conserved. Thank you for your advice, and we have indicated these sites in figure 2.

  1. For the WB analysis, why the band of TRE1 in figure 8D is blue? Usually, the WB results show black bands, and only the prestained molecular weight marker is colorful. Please double-check your results shown here. And the band in figure 8c is wider than that of 8A, please uniform the scale.

Response: At present, there are antibodies (secondary antibodies) with specific IgG. The secondary antibodies can bind various markers. The most commonly used one is enzyme-linked secondary antibodies. The enzymes used in enzyme-linked antibodies are usually alkaline phosphatase (AP) or horseradish peroxidase (HRP). Alkaline phosphatase can convert the colorless substrate 5-bromo-4-chloroindolyl phosphate (BCIP) into a blue product. In this study, we chose AP-conjugated Goat anti-mouse IgG as the second antibody. So we will found the bond of TRE1 protein in figure 8D is blue.

The enhanced chemiluminescence method is used to detect horseradish peroxidase. In the presence of H2O2, horseradish peroxidase oxidizes luminol and emits light. The presence of horseradish peroxidase can be detected by placing the imprint on the photographic negative. So usually we saw the WB results show black band.

Considering the Reviewer’s suggestion, we have uniformed the scale of the picture in figure 8c and 8A.

  1. The Chinese words in line 301 and 302 should be deleted.

Response: We apologize for the format problems in the original manuscript. We have deleted the Chinese words in the manuscript.

15.For the RNAi part, when were the adults fed with dsRNA, day 1 or day 2?

Response: The adults fed with dsRNA on the day 1. First the dsRNA feeding was continued for 12 h on day 1,then the leaves were replaced with fresh rice leaves without dsRNA every 24 h. 

  1. In figure 9, what’s the amount of purified TRE1 used for these analyses?

Response: The amount of purified LoTRE1 used for these analyses were 10 µl. The reaction mixture (1 ml) consisted of 10 µl purified protein, 50 µl trehalose (200 mmol/L), and 940 µl PBS.

  1. Line 369, please remove the blank.

ResponseWe are very sorry for our incorrect writing. We have removed the blank in the manuscript.

  1. Line 399, What’s the Ai in “Ai revealed”?

ResponseWe apologize for the language problems in the original manuscript. We re-wrote the sentence in the revised manuscript as the following: Ai et al. (2018) revealed that the optimum temperature of the trehalase from Helicoverpa armigera was 55 ℃.

  1. Line 418, Ha Treh or HaTRE?

ResponseWe apologize for the language problems in the original manuscript. We have changed “HaTreh” to “HaTRE1”.

  1. Line 472, Bombyx mori should be black.

ResponseWe apologize for the language problems in the original manuscript. We have blacked the sentence of Bombyx mori.

Once again, thank you very much for your constructive comments and suggestions which would help us both in English and in depth to improve the quality of the paper.

Thank you and best regards,

Yours sincerely,

Qingtai Wang

This manuscript is a resubmission of an earlier submission. The following is a list of the peer review reports and author responses from that submission.

Round 1

Reviewer 1 Report

The authors describe the isolation of the Trehalase gene 1 (LoTRE1) encoding the soluble form of the protein from the rice water weevil Lissorhoptrus oryzophilus, the bacterial overexpression and purification of the protein and its characterization regarding temperature and pH optimum for the enzymatic activity. Furthermore, the authors compare the putative 3D structure of the protein with the crystallized E.coli protein and investigate the putative interaction of the protein with its substrate Trehalose, using different bioinformatics prediction tools. Finally, they knock down the expression of the protein using loTRE1-dsRNA fed to adults and investigate the effect on insect survival and LoTRE1 mRNA levels.

The manuscript in its current state is not acceptable for publication. Major revision is required before this manuscript can be considered for publication.

The whole manuscript needs a thorough revision of the English language, paying attention to word choice and correct description of facts, referencing within sentences, use of singular/plural, use of the definite article, use of the correct tense. Moreover, the authors often use vague terms to describe something instead of simply using the specific name of something they are talking about. I highly recommend the authors to have their manuscript checked by a language expert before resubmission.

Just a few examples of major language issues from the first page:

11: As a quarantine pest, it is widely distributed and seriously damaged in China – it might cause serious damage in China, but the insect is certainly not seriously damaged

12: in this research study we have cloned one soluble trehalase gene. -  the gene is not soluble; the gene product is soluble. So it should rather be: we cloned one trehalase gene encoding the soluble protein version. 

23: In this report, we gift a basic study – “gift”? 

19: Cause the important role of trehalase in insects, this experiment explored the molecular characteristics andgene function of soluble trehalase of rice water weevil – not understandable!

 27: Heterologous expression of LoTRE1 in L. oryzophilus indicated that the recombinant protein 27 has the ability to decompose trehalose – the protein was not heterologously expressed in L. oryz. but in E.coli. and the heterologous expression does not indicate that the protein has the ability to decompose trehalose. The experiments done with the heterologously expressed and purified protein indicate this.

Moreover, the methods section needs a thorough revision and rewriting in large parts. The way the methods are described here the experiments can’t be replicated. A lot of specific information is missing and descriptions are often cryptic, leaving the reader to guess what might have been done. Even though I picked out a lot of points, the list below is not complete.

Finally, results and discussion also need revision. Please see all my specific comments below.

Specific comments:

Simple summary:

Needs complete rewriting and English correction

Introduction:

44-47: more reference are needed here to support all the functions of trehalose listed here. The two cited ones don’t cover this

47-53: it should be made clear here that insects don’t have blood, but hemolymph, which can be considered the blood of insects

56: update the list and literature here. E.g Neyman et al 2021

56-57: if something is known about the function of the elements the authors should include this information here, and also talk about the function of the signal peptides for the non-expert reader

61-62: reference missing

63: The common name of L. oryz. – rice water weevil - should be introduced here, as it is used later throughout the manuscript.

69-70: the authors could elaborate a bit more here on why trehalose metabolism might be relevant for invasive potential. Other studies on this?

65-76: having invaded temperate growing areas around the world includes Europa and Asia. Unless the latter areas are not temperate. Then this should be specified here.

72-82: these two paragraphs are redundant

77: the function is already known, it cleaves trehalose; the authors studied the effects of knocking down the gene by RNAi on the survival of the beetle

83: understanding the perform of--- English!

The authors refer in the abstract to Asp333 being obviously a key amino acid in the active center, and they model active center interactions with the substrate, and compare that to other insects in the discussion but don’t talk at all about what is known about the catalytic mechanism and active center structure and key amino acids of trehaloses and specifically insect trehaloses in the introduction. This important topic for this manuscript should be included and described well here

Materials and methods:

2.1

89 – fresh rice? Plants? Collected as adults or larvae?

99 -  how much RNA used for cDNA reaction?

2.3

103 -  how was the LoTRE1 sequence identified? Analytical/bioinformatics specifics?

104 – sequences? Was there more than one?

Settings of the different programs? These analyses can’t be repeated without specifics about the program settings

2.4

124 - Has GAPDH been verified as a suitable ref gene before? Stable expression across all tissues? if so explain when and how (own work or cite); in not suitability of GAPDH as ref gene in Lo needs to be verified

125 -  green? = SYBR green?

138: the boxes are not only the restriction sites, there are additional bases included in the boxes

2.5

143 – monoclones were checked? What does that mean – checked? Specify

147 – respectively implies that some of the IPTG concentrations where used at one temperature, some at the other. I think this is not correct here, and if it is correct then it’s not clear which ones where used at which temperature.

149 – purify the supernatant. The purity was checked – purify what? Purity of what?

2.6

152 – sample = what? Transfer conditions?

153-157 – specifics of the blocking and antibody incubations: temperature, concentrations; source of the AB?

158 – BCIP/NBT substrate solution? – is that the official name? if not give official name and manufacturer

2.7

161 – to indirectly firm?

161 – I understood the trehalase activity was investigated according to 2.9? what’s the difference between 2.7 and 2.9? 

166 - DNS acid method – kit used? Manufacturer

167 – the unit definition stems from the authors or from the kit? Or is this meant to say that a general definition was used?

168 – what are the standard assay conditions? They are not defined anywhere?

170 – substrates? Be specific

173 -  three times with enzyme from the same purification? Or different bacterial culture and purification?

2.8

176 – ground on …  language!

179 – 180: divided in 3 groups and each group consistet of 3 biol replicates  confusing; and how large were the groups then in the end? I guess 30, whoch is also mentioned later, but the description isn’t clear or useful.

181 – both ends of fresh leaves: only the ends of the leaves were used for that? Why?

182 -184 – size of tubes used, what are EP tubes?

186 – rice leaves were replaced every 24 h: but how long did the exposure to the dsRNA last in total?

Completely missing: experimental setup for the qPCR analysis of the dsRNA feeding: where did those samples come from? From a different, independent experiment? Or the animals that survived the dsRNA feeding? How many animals were sampled for one qPCR biological replicate? How many replicates (biological and technical)?

2.9

190 – trehalase assay kit: does not exist – at least can’t find a kit named like this from the indicated vendor. Please specify

How many insects were used for one biological replicate? Where did these individuals come from? From the 2.8 experiment? Or were independent dsRNA feeding assays conducted for the trehalase activity and trehalose content assays?

2.10

197 – there is no significant and extremely significant in statistics

Results:

3.1:

This chapter needs complete rewriting. The data presented here is incomplete and incomprehensive

Was only one trehalase gene identified?

201 – how was the ORF identified? The authors don’t specify at all on which bioinformatics analysis and database this is based. Based on the methods section (2.3) I assume transcriptome data were analyzed. But for what? Was it found by homology search? If so, then by using which reference organism? Please specify how the ORF was searched and what the results were.

203 – which signal peptide was predicted? And based on what? What does it mean for the function of the protein or its expression?

204—205: what are these specific motifs? What is their biological relevance? 

206 – TMHMM provided a prediction that there are no transmembrane domains.

217 – I assume the authors want to say based on the amino acid sequence

Figure 1: I believe that the position indicated for the G-rich region is not correct. It should be shifted to the left. Moreover, the figure size is quite small. Increase the size for readability

218 – name the ones the LoTRE1 was compared to and at least also name the one with the lowest homology

Figure 2: line 232 – with specific motifs the authors mean the signature motifs? If yes then they should use this term here, if not they have to explain what kind of motifs they are talking about here and how they were identified and why/what their relevance is

233 – the white amino acids in the alignment are not conserved! Grey ones are conserved and black ones highly conserved

Where is the G-rich motif? Indicate in the alignment

3.2: where also other tissues probed besides the ones shown in Fig. 4? If so then they should be included here. Was also the complete insect probed? Which of the samples served as a reference (i.e. expression level set to 1, and the other samples compared to it)?

3.3:

254 – in the best condition: that’s not comprehensible, the authors should use more accurate language for describing the results of the Ramachandran plot.

256 - template macromolecule: I assume the authors mean the E.coli protein? They should use a specific description here

260-61 – contrasts are highlighted with arrows? I can see only one

263 – Ramachandran plot

266 - the repaired 3D protein? I believe something is wrong here?

269 – which other compounds?

3.4

276-277 – same as in the methods. I believe the use of the word “respectively” is not correct here.

281 – it was expressed mostly as soluble protein, not “in the form of supernatant”; aside from the language issue I can’t follow the statement that it was mostly expressed in the soluble form, as the band from the pellet/insoluble fraction is also quite strong

Figure 8: which marker was used? The expected position of the Trehalase in gel A should be indicated by an arrow.

Overall, the figure should be larger

288-292 – not the protein is induce, but the protein production in the bacterial cells

292-294 – change ultrasonic crushing to sonication 

3.5

Was the pH optimum determined before the temperature optimum or vice versa? At which temperature was the pH dependency measured? And at which pH was the temperature optimum determined? If the pH optimum was only probed at one temperature and the temperature optimum only probed at one pH, how do the authors know that different pH will not have an influence on the pH temperature optimum or vice versa? This should be tested

300 – at temperatures above 75°C

302 - specific the optimal conditions; how was the specific activity determined?

303 -  how were the kinetic parameters determined?

Figure 9: If the temperature was done first and then the pH at the optimal temperature, then A and B in Figure 9 should be switched.

Legend (9c) – specify the units of the y and x axis

3.6

310 -  was it the authors’ intention to show that RNAi in general is functional in L. oryz.? or was it the authors intention to investigate how essential loTRE1 is for the insects? In this case they should rephrase this

Figures: Increase size and readability

Discussion:

General: the authors should make clear if they are discussing their own work or that of others by naming the species. E.g. 363-364: tissue expression pattern analysis revealed that in L. oryz….., 

346: The resultant Ramachandran plot of the LoTRE1 model agreed with the requirements for docking – what does that mean?

353: Moreover, in docking studies, the best position for LoTRE1 was obtained with a Glide XP score of −1.36 kcal/mol – what does that mean? Not comprehensible for an non-expert on molecular docking

356-57: There were differences regarding the ASP residue as a proton donor for animal trehalase [26]. – unspecific! Animal?

362-363: incomprehensible!

376: apis mellifera

380-82: system temperature - Which system?

385-86: D. melanogaster thrives in temperate regions. How would this explain the higher temperature tolerance of the D. mel. enzyme??? The authors should come up with a better hypothesis. What is known about the role of the high temperature optimum (around 50°C) of this enzyme in general? None of the insects lives at such high temperatures. 

390: in adults insects! Why where only adults tried, not larvae?

393: a dramatic decrease is not scientific. And 50% is significant, but still leaves 50% of the enzyme. Correspondingly, only 12% adults died. So obviously, the adults do pretty well with only 50% of the protein. Which will not qualify RNAi against LoTRE1 as a suitable instrument for pest control!

390-402: it is not clear how the authors get from their RNAi results (death of a few insects) to the hypothesis, that loTRE1 is involved in L. oryz. chitin production. Based on the diverse roles of trehalase in insect metabolism, this could have many different reasons. This connection (if existing) has to be much better explained. Otherwise, this is not a hypothesis based on results and literature, but phantasy and should be deleted.

406: no, the observed RNAi effect (death of 12% of insects) did not at all infer the function of trehalase. 

410: as already said: killing 12% by RNAi will not qualify RNAi against LoTRE1 as a suitable instrument for pest control!

General:

Would be interesting to include 3D structure model comparison with other insect trehalases as additional figure, like Shukla 2018, Neyman 2021, Ai 2019

Update current literature and compare the study to this literature, e.g.:

Neyman et al; The Protein Journal https://doi.org/10.1007/s10930-021-10032-7

Review on insect trahlases by Shukla et al. Glycobiology 2015

Reviewer 2 Report

This paper is a conventional study in the field of gene function. The fact is that the basic function of soluble trehalase has been studied too thoroughly. In this context, we expect a new finding or viewpoint. While no treatment was set up to address some specific issues according to this manuscript. To cope with this issue, we think the section of Introduction and Discussion must be substantial revised to highlight the value of this study. In addition, if necessary, the workload needs to be increased (not limited to just one gene), or stress treatments can be introduced (Trehalose is closely related to insect stress resistance). Also, we think this study seems to not match the topic about environmentally-Friendly pest Control approaches for invasive insects.

Specific issues:

1. In Abstract, the description it also led to 12% death is inaccurate.  LoTRE1 is indispensable for... is not detailed enough to explain the significance of this study. Purpose of the study feels lightweight and unconvincing in the abstract.

2. Each paragraph in the Introduction is isolated. Line 72-85: the two paragraphs seem to be repetitive contents.

3. In Figure 10A, silencing efficiency for 24 h should be present in this figure.

4. In Figure 10B, the significant differences between dsGFP and RNase free water groups is a error.

Reviewer 3 Report

The present contribution aims to characterize the structure and function of trehalase from the insect L. oryzophilus. This is an interesting topic and the present paper uses the adequate strategy to deal with that problematic. However, I would add several points in the paper before it can be accepted.

1) For the kinetics studies, the experiments were done at 50°C for 30 min. The authors must ensure that the trehalase doesn’t denature during that time at that temperature.

2) Putative glycosylation sites are highlighted. Authors should add a discussion about their potential importance. They also should comment that the trehalase, which is overexpressed in E coli, don’t have these glycosylation sites.

3) The quality of figures 1 and 2 should be revised. They are too small.

4) In the multiple amino acids alignment, one sequence of the H axyridis trehalase was chosen. In H axyridis, 5 isoforms are known. Authors should justify their choice.

5) The purpose of the docking studies should be a bit more justified. Authors are doing these studies but nothing is actually done from the data obtained.

In addition, there is a confusion about the docking method used. In the discussion, the Glide XP score is referenced while Autodock is mentioned in the Materials and Methods.

6) The obtained Km is relatively high. The authors should comment and compare to a recent paper (see below) showing the same range of Km.

Neyman V, Francis F, Matagne A, Dieu M, Michaux C, Perpète EA. Purification and Characterization of Trehalase From Acyrthosiphon pisum, a Target for Pest Control. Protein J. 2022 Feb;41(1):189-200.